

# Therapeutic strategy and prognostic analysis of inflammatory myofibroblastic tumor in the head and neck: a retrospective study

Feng Liu[1,*], Yanchao Qin[1,*], Zhiwei Zhang[2], Mengru Li[3], Bowei Feng[4], Wei Ding[1] and Shubin Dong[1]

[1] Department of Head and Neck Surgery, Shanxi Provincial Cancer Hospital/Shanxi Hospital Affiliated to Cancer Hospital, Chinese Academy of Medical Sciences/Cancer Hospital Affiliated to Shanxi Medical University, Taiyuan, Shanxi, China
[2] First Clinical Medical School, Shanxi Medical University, Taiyuan, Shanxi, China
[3] Academy of Medical Sciences, Shanxi Medical University, Taiyuan, Shanxi, China
[4] School of Stomatology, Shanxi Medical University, Taiyuan, Shanxi, China
[*] These authors contributed equally to this work.

## ABSTRACT

**Objective**. This study aimed to investigate the clinical features, treatment methods, and prognosis of head and neck inflammatory myofibroblastic tumor (HNIMT).

**Methods**. A retrospective analysis was conducted using the clinical data of 12 HNIMT patients who were admitted to Shanxi Cancer Hospital between January 2016 and December 2023. This analysis focused on their clinical manifestations, pathological characteristics, treatment strategies, and prognosis.

**Results**. Among the 12 cases analyzed, four involved inflammatory myofibroblastic tumors (IMT) located in the nasal sinuses or nasopharynx, with symptoms including nasal congestion, rhinorrhea, and maxillofacial swelling. Two cases each in the salivary glands and oral cavity presented as localized, painless masses. One right cervical IMT case also presented as a painless lump. Two laryngeal IMT cases had hoarseness, and one subglottic endotracheal IMT case showed inspiratory dyspnea. All patients received surgery, with postoperative pathology confirming IMT. During follow-up, four cases recurred. Finally, nine patients were disease-free, two survived with disease, and one died.

**Conclusions**. HNIMT is a rare, low-grade malignant or borderline tumor that is generally associated with a favorable prognosis. Accurate diagnosis relies on pathological examination, and surgical resection remains the primary treatment for HNIMT. The need for adjuvant therapy following surgery should be determined by clinicians based on tumor location, surgical approach, and the presence of high-risk factors.

Corresponding author
Shubin Dong, 110669095@qq.com

## INTRODUCTION

Inflammatory myofibroblastic tumor (IMT) is a rare mesenchymal soft tissue tumor. It is characterized by histological features of both neoplastic and inflammatory components in the same lesion, and involves both myofibroblastic and inflammatory cells. It is typically classified as a low-grade malignant or borderline tumor (*Zhang et al., 2020a*; *Zhang et al., 2020b*). IMT primarily affects children and adolescents, although it can also occur in adults. The exact etiology and pathogenesis of IMT remain poorly understood (*Al Shenawi et al., 2022*; *Camela et al., 2018*). This tumor can develop throughout the body, with the lungs being the most common site, followed by the mesentery, omentum, abdominopelvic organs, head and neck, limbs, and breasts (*Siemion et al., 2022*; *Mahajan et al., 2021*). IMT has been documented in various regions of the head and neck including the orbit, nasal sinuses, salivary glands, larynx, and thyroid gland (*Baranov & Hornick, 2020*; *Korlepara et al., 2017*; *Kim et al., 2016*). Due to the complex anatomical structure of the head and neck, clinical manifestations of IMT can widely vary and often lack distinctive symptoms or signs. Systemic manifestations and imaging features of IMT are also atypical. Due to these factors, clinically distinguishing IMT from other malignant tumors before surgery may be challenging (*Kaytez, Kavuzlu & Oguz, 2021*; *Zhang & Liu, 2023*; *Kerr et al., 2021*).

The incidence of head and neck IMT (HNIMT) is low, and most studies on this disease have been restricted to individual case reports or small case series. Consequently, research on the clinicopathological features, treatment strategies, and prognosis of HNIMT is still limited. This study presents a retrospective analysis of 12 patients with HNIMT, summarizing clinicopathological features, treatment approaches, and outcomes. The study's goal was to improve the understanding of this disease and provide valuable insights into the standardized diagnosis and management of HNIMT.

## MATERIALS AND METHODS

### Clinical data

The clinical data of 12 patients with HNIMT who were admitted to Shanxi Cancer Hospital from January 2016 to December 2023 were collected. All patients were diagnosed with IMT by pathology and had complete medical records and follow-up data. Informed consent was obtained from all patients, and the study was reviewed and approved by the Medical Ethics Committee of Shanxi Cancer Hospital (Ethics number: KY2023231).

### Inclusion and exclusion criteria

Inclusion criteria:
(1) Patients with a pathological diagnosis of HNIMT;
(2) Patients with complete clinical data;
(3) The follow-up period was >6 months.
Exclusion criteria:
(1) Patients with incomplete pathological data;
(2) Patients with missing clinical data;
(3) Patients with other malignancies;
(4) Patients who cannot be followed up effectively.

### Diagnostic criteria for IMTs

Pathology is the "gold standard" for diagnosing IMTs. We examine all tissue samples by hematoxylin and eosin staining (HE) and immunohistochemistry (IHC), with some patients also undergoing molecular testing. The tests include smooth muscle—specific antibodies (SMA), vimentin, muscle-specific actin (MSA), ALK, CD117, CD34, and S-100. In the study, we conducted pathology examinations on all samples according to the 2022 WHO classification criteria to meet the diagnostic standards for IMTs.

### Treatment strategies

Treatment options were determined through multidisciplinary team (MDT) discussions and were based on the patients' clinical symptoms, tumor stage, tumor location, the tumor's relationship to surrounding tissues, and the overall condition of the patient.During the operation, frozen section pathology was routinely performed to determine the extent of surgical resection and to ensure negative surgical margins. Adjuvant therapy is recommended for high-risk patients (such as those with positive surgical margins, postoperative recurrence, or histological evidence of malignant transformation) to reduce the risk of recurrence and improve survival rates. The implementation of the relevant treatment plans was carried out with the informed consent of the patients and their families, and an informed consent form was signed.

### Follow-up

In the first year following surgery, patients attended follow-up visits at the outpatient clinic every three months. Starting from the second year post-surgery, regular check-ups were arranged at six-month intervals. Re-examinations encompassed both imaging and laboratory examinations. Throughout the follow-up period, disease recurrence was confirmed through imaging or histopathological examination. For those unable to come to the hospital for follow-up appointments, telephone follow-ups were conducted and recorded the relevant follow-up results.

## RESULTS

### Clinical characteristics

The study included a total of 12 patients with seven males and five females. The age of HNIMT onset varied from 7 to 72 years, with an average age of 48 years. The lesions and clinical manifestations observed in these patients were as follows: Three cases involved the nasal cavity and sinuses, and one case affected the nasopharynx. Symptoms for these cases included nasal congestion, rhinorrhea, and maxillofacial swelling. There were two cases of salivary gland (specifically parotid gland) IMT and two cases of oral cavity IMT (including the tongue and gums), as well as one case of right cervical IMT, all of which presented as a localized, painless mass. Two cases involved the larynx and presented as hoarseness, and one case in the subglottic trachea manifested as inspiratory dyspnea. The average time from the onset of clinical symptoms or tumor mass detection to diagnosis was 35 days (range: 10–180 days). No distant metastases were detected in any of the patients following systemic examination. The general clinical data of the patients are shown in Table 1.

**Table 1  Clinicopathologic characteristics of HNIMT patients ($n = 12$).**

| Basic characteristic | $n = 12$ |
|---|---|
| Median age (years, range) | 48(7–72) |
| Gender (male:female) | 7:5 |
| Tumor location ($n$, %) | |
|     Nasal cavity and paranasal sinuses | 3 (25.0) |
|     Nasopharynx | 1 (8.33) |
|     Larynx 2 | (16.66) |
|     Subglottic endotracheal | 1 (8.33) |
|     Parotid gland | 1 (8.33) |
|     Submandibular gland | 1 (8.33) |
|     Tongue | 1 (8.33) |
|     Gingiva | 1 (8.33) |
|     Right neck | 1 (8.33) |
| Clinical manifestation ($n$, %) | |
|     Nasal congestion, running, and swelling of the maxillofacial region | 4 (33.33) |
|     Painless masses | 5(41.67) |
|     Hoarseness | 2 (16.67) |
|     Dyspnea | 1 (8.33) |
| IHC ($n$, %) | |
|     SMA (+) | 10 (10/12) |
|     Vimentin (+) | 8 (8/ 12) |
|     Desmin (+) | 7 (7/10) |
|     ALK (+) | 6 (6/11) |
|     MSA (+) | 5 (5/8) |
|     S-100(−) | 7 (7/9) |
|     Dog-1(−) | 5 (5/7) |
|     CD117(−) | 8 (8/9) |
|     Ki-67 | 10 (3%–30%) |
| Status at last follow-up | (n, %) |
|     Non-neoplastic death | 1 (8.33) |
|     With disease survival | 2 (16.67) |
|     Disease-free survival | 9 (75.0) |

## Imaging features

Ultrasonographic imaging showed that HNIMT often exhibited homogeneous or heterogeneous solid masses with well-defined or infiltrative borders. HMIMTs are mostly demonstrated as solid nodules on US images, and a few may be appeared as cystic-solid nodules with punctate blood flow signals within the lesions. Two cases of salivary gland IMT presented as hypoechoic, irregularly-shaped solid masses with sparse blood flow. IMT of the soft tissue in the right neck showed hypoechoic nodules in the sternocleidomastoid muscle with well-defined borders. The findings of the parotid IMT ultrasound analysis are shown in Fig. 1.

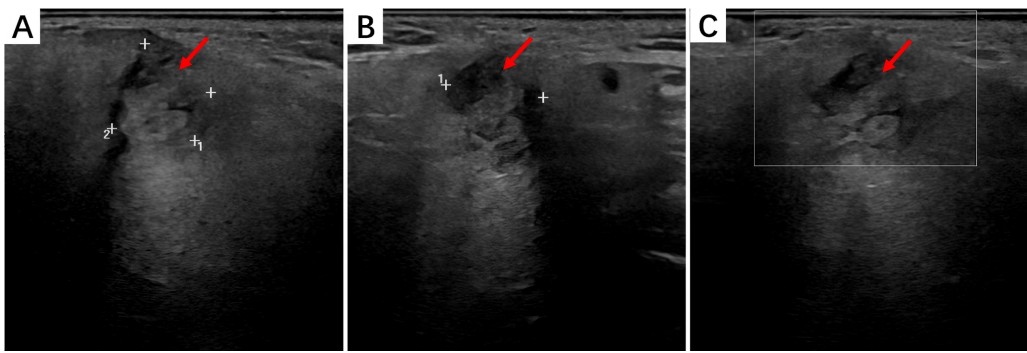

**Figure 1** **The ultrasound findings of right parotid IMT.** (A) A mass can be seen in the superficial lobe of the right parotid gland with unclear boundaries and irregular shape; (B) The mass has no obvious capsule, and most areas show echogenicity without obvious calcification or liquefaction; (C) CDFI detection shows no obvious blood flow signal.

Electronic endoscopy is primarily utilized for examining lesions in the nasal sinuses, nasopharynx, and larynx.During endoscopic examination, normal mucosa, polyps, edematous mucosa, hypertrophic rhinitis, or bloody rhinorrhea were observed in nasal sinus lesions.In this study, smooth, cyst-like masses on the vocal cords were seen in laryngeal IMT cases, while fragile neoplasms with purulent or bloody secretions were observed in nasal sinus IMT cases. Subglottic tracheal IMT, located one cm below the glottis, caused an approximately 70% narrowing of the tracheal lumen. The laryngeal IMT endoscopic presentation is shown in Fig. 2.

Due to the anatomical structure of the head and neck, HNIMT mainly presents on computed tomography (CT) scans as an expanding, round, or irregular lobulated soft tissue mass, which can easily invade the surrounding soft tissue and bone structures and manifest as local bone erosion. Tumor presentation can differ in images with contrast-enhanced scanning. For example, solid sections may show progressive or uneven enhancement. Non-enhanced cystic necrosis areas and calcification lesions can also be observed in some lesions. In this study, nine patients underwent CT imaging, and six of these patients underwent contrast-enhanced scans. Contrast-enhanced CT images of the nasal cavity and paranasal sinuses in two cases showed moderate enhancement with unclear boundaries, accompanied by compressive bone resorption in adjacent bones. Non-contrast CT images of IMT lesions in the subglottic trachea demonstrated flaky, slightly low-density shadows on the right posterior wall of the subglottic trachea, part of which protruded into the tracheal lumen with unclear boundaries. The tongue lesion was significantly enhanced on contrast-enhanced CT images and remained so during the delayed phase. No enhancement was observed on contrast-enhanced CT scans of the nasopharynx and right neck soft tissue lesions. The CT manifestations of IMT in the nasal cavity and paranasal sinuses are shown in Fig. 3.

On magnetic resonance imaging (MRI) scans, HNIMT typically demonstrated as nodular signal shadows with irregular shapes and unclear boundaries. T1-weighted imaging (T1WI) showed mixed but primarily low signals, and T2-weighted imaging (T2WI) showed uneven
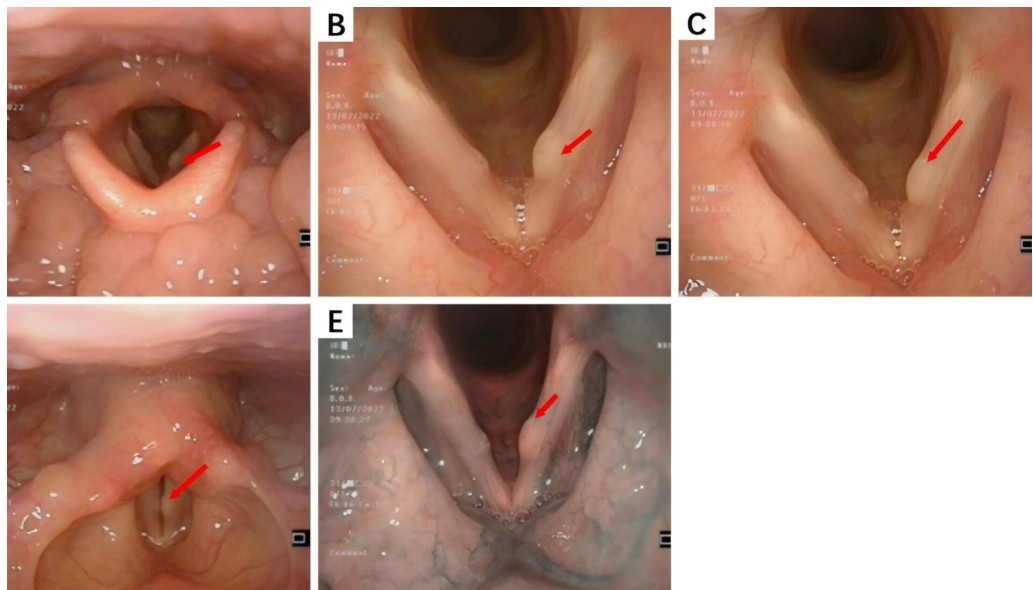

**Figure 2  The endoscopic findings of laryngeal IMT.** (A) A cyst-like mass can be seen in the anterior middle segment of the left vocal cords; (B, C) The surface mucosa of the mass is smooth; (D) Bilateral vocal cords have good activity; (E) Narrow Band Imaging (NBI) model: Intraepithelial Papillary Capillary Loop (IPCL) is almost invisible, with a clear course of oblique and dendritic blood vessels and no significant increase in diameter.

and slightly high signals. The extent of enhancement is closely related to the components within the lesion. In this study, MRI examination was performed on five patients, with three of them undergoing contrast—enhanced scanning. Non-contrast MRI of laryngeal IMT revealed a nodular shadow in the right front part of the glottis area, showing a slightly prolonged T2 and T1 signal. Contrast-enhanced scanning significantly enhanced this lesion, blurring its base. Abnormal signals at the left front edge of the tongue were seen in the image of non-contrast MRI, and the lesion was significantly enhanced during contrast-enhanced scanning. Based on the above imaging findings, the lesion could be a benign hemangioma. Non-contrast MRI of the gingiva IMT showed an oval-shaped long T1 and long T2 signal in the right gingiva and buccal region. The contrast-enhanced MRI showed obvious heterogeneous enhancement, with a non-enhanced area in the center. The boundary between the lesion and the surrounding structures was clear, and the adjacent bone was not involved. The MRI manifestations of tongue IMT are shown in Fig. 4.

In summary, three patients underwent neck ultrasound, with the results indicating low-grade tumors. Nine patients underwent CT scans and five patients underwent MRI scans. Contrast-enhanced imaging demonstrated different enhancement degrees in various lesions. These results suggest difficulties in establishing an accurate diagnosis based solely on preoperative radiological assessments.

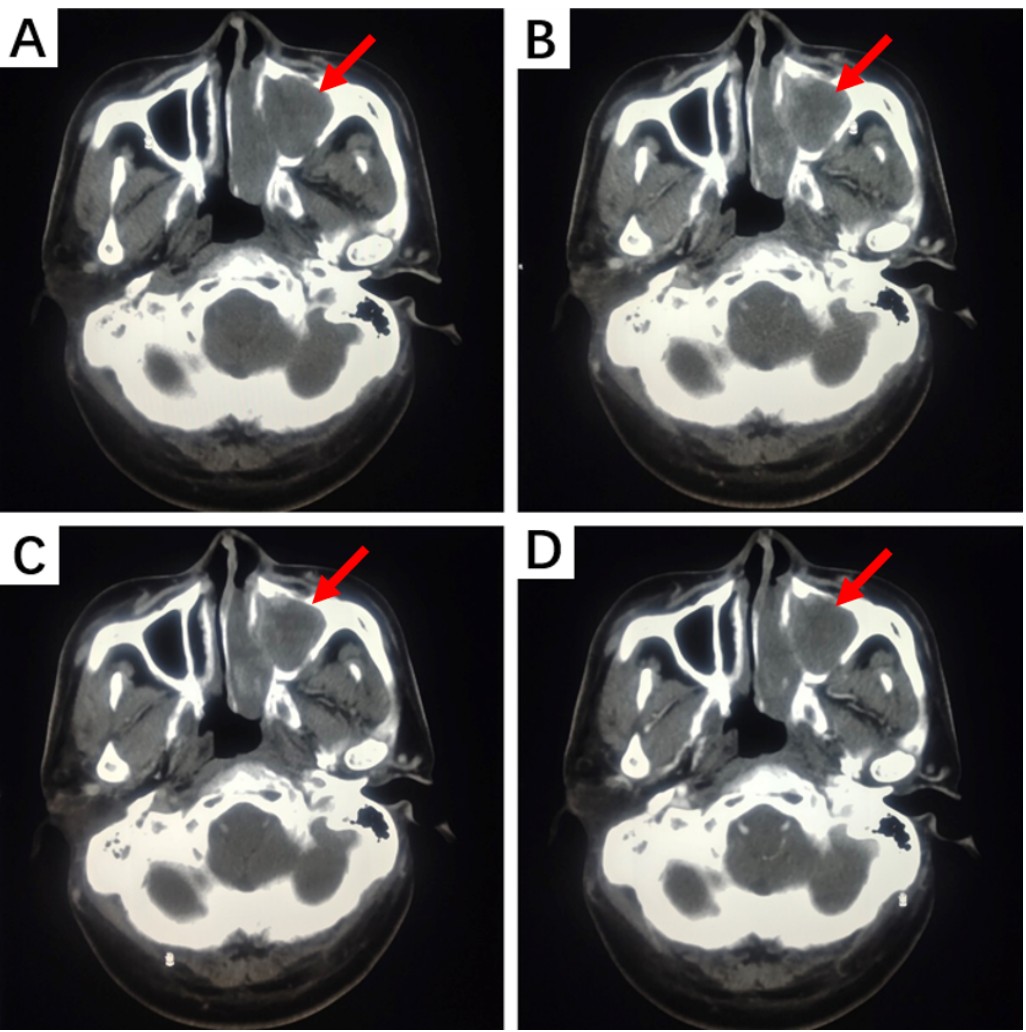

**Figure 3** **CT examination of a patient with left nasal cavity and paranasal sinus IMT.** (A) Non-contrast CT scan shows irregular soft tissue density shadows filling the left nasal cavity and maxillary sinus, with surrounding bone thinned and interrupted. The nasal septum is compressed and bent to the right. (B) Contrast-enhanced CT scan during the arterial phase reveals uneven and moderate enhancement. (C) During the venous phase, further enhancement is observed. (D) In the delayed phase, enhancement slightly decreases.

## Pathological features

At a macroscopic level, most IMT cases presented solid lesions with a hard texture, clear boundaries, and off-white or grayish-yellow surfaces. Typical macroscopic appearances are shown in Fig. 5. At a microscopic level, the tumors exhibited spindle- or short-spindle-shaped myofibroblastic cell proliferation with mild atypia and occasional mitoses. Varying degrees of myxoid degeneration, hyaline changes, and collagen formation were also observed in the stroma, along with infiltration by lymphocytes and plasma cells. Based on differences in cell proportion and distribution, six cases were classified as mucinous/vascular type, two cases as spindle-cell dense type, and four cases as fibrous

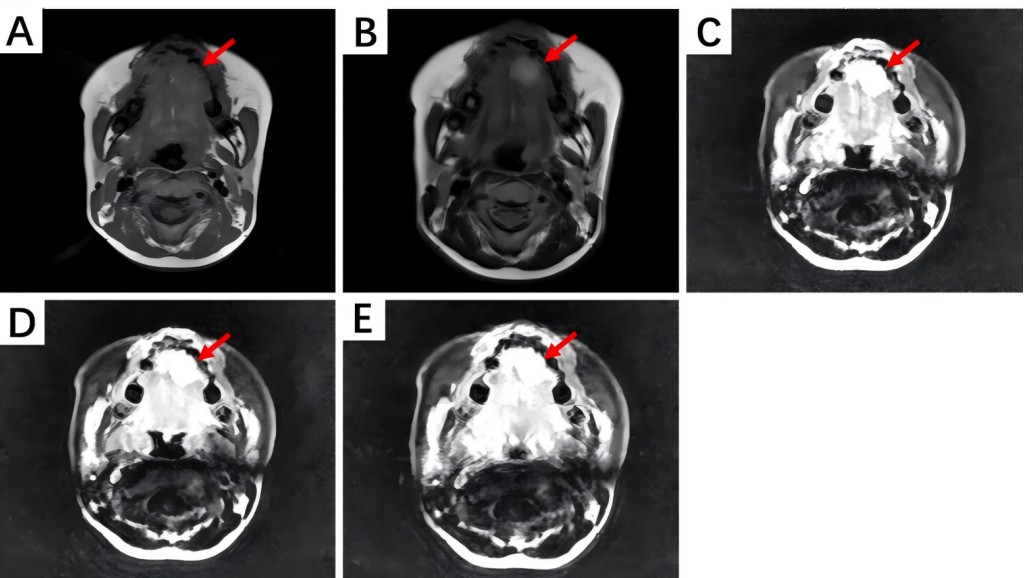

**Figure 4** **The MRI manifestations of tongue IMT.** (A) Non-contrast T1-weighted imaging shows an iso-intense signal lesion on the left tongue edge. (B) Non-contrast T2-weighted imaging shows a hyper-intense signal. (C) Contrast-enhanced scan during the arterial phase reveals marked enhancement. (D, E) Contrast-enhanced scans during the venous and delayed phases show persistent enhancement.

type. IHC staining showed that 10 cases tested strongly positive for SMA, eight cases tested strongly positive for Vimentin, seven cases tested positive for Desmin, and five cases tested focally positive for MSA. S-100, CD117, and Dog-1 were often negative, and the positive rate of Ki-67 was 3%–30%. Typical HE staining and IHC are shown in Fig. 6. Six cases tested positive for Anaplastic lymphoma kinase (ALK), and two cases underwent fluorescence *in situ* hybridization (FISH) testing. Detailed results of the IHC analysis are displayed in Table 2.

## Treatment and survival

All 12 study patients underwent surgical treatment, with three receiving adjuvant radiotherapy after surgery. Four patients experienced local recurrence, and all of these patients underwent a subsequent surgical resection. Postoperative treatments for IMT in the maxillary sinus and nasopharynx included hormone therapy combined with ALK inhibitor therapy and either chemoradiotherapy or chemotherapy. The specific treatment and follow-up data are shown in Table 3.

Patients with IMT of the parotid gland underwent two operations. The first operation was the resection of the superficial parotidectomy and mass. Postoperative pathology revealed that the tumor had no capsule and had invaded the parotid tissue and surrounding muscle, with a Ki-67 index of about 30%. Without postoperative adjuvant therapy, the tumor recurred four years after the initial surgery. Consequently, the patient underwent a right parotidectomy. At the last follow-up, the patient was recovering well, with a disease-free survival status of 30 months after the second surgery.

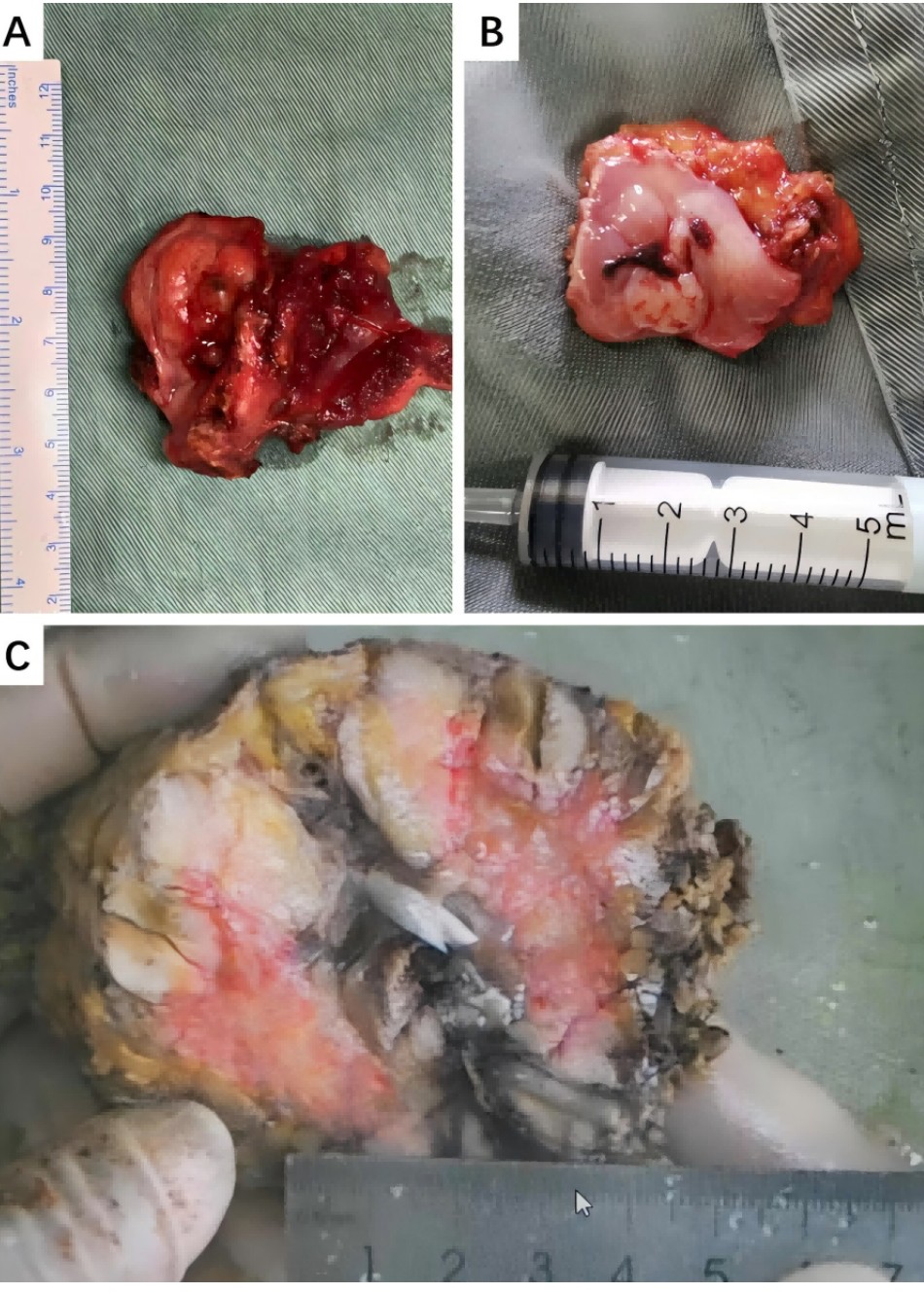

**Figure 5** **Typical macroscopic appearances of HNIMT.** (A) Right cervical IMT, the cut surface of the tumor is grayish-yellow or grayish-red with clear margins.; (B) Gums IMT, the surface of the tumor is ulcerated and has a cauliflower-like appearance; (C) Right submandibular gland, the cut surface of the tumor is grayish-white, with areas of cystic and necrosis.

The patient with IMT of the left maxillary sinus also underwent two operations. The first procedure involved a lateral nasal incision and a subtotal maxillectomy, and the patient experienced a satisfactory postoperative recovery and no adjuvant therapy. However, two

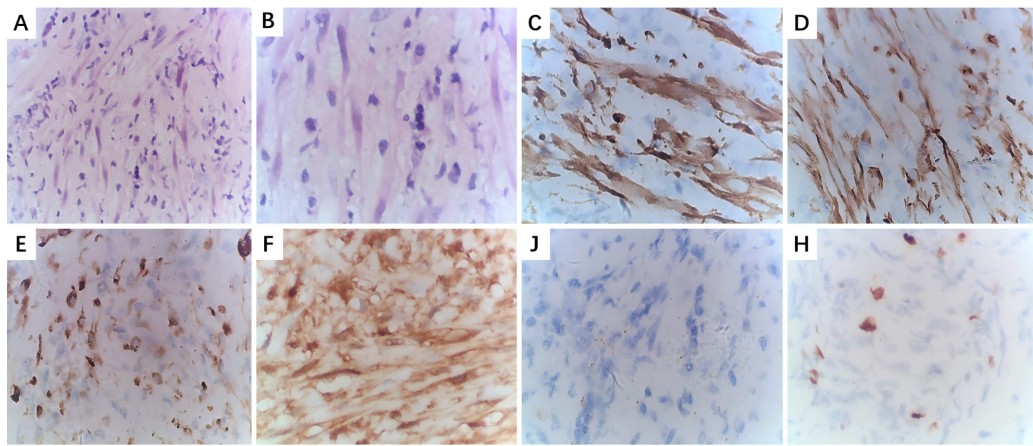

**Figure 6   Pathological examination of larynx IMT.** (A): HE staining (×100); (B) HE staining (×400); (C) IHC staining for ALK (+) (×400); (D) IHC staining for SMA (+) (×400); (E) IHC staining for Desmin (+) (×400); (F) IHC staining for S-100 (−) (×400); (G) IHC staining for CD117 (−) (×400); (H) IHC staining for Ki-67, approximately10% (×400).

**Table 2   IHC characteristics of 12 HNIMT patients.**

| Case | SMA | Vimentin | Desmin | ALK | MSA | CD117 | S-100 | Dog-1 | Ki-67 |
|---|---|---|---|---|---|---|---|---|---|
| 1 | + | − | + | NP | NP | − | − | − | 3% |
| 2 | + | + | weakly+ | + | + | − | NP | NP | 10% |
| 3 | − | + | NP | + | weakly+ | NP | − | weakly+ | 30% |
| 4 | + | + | − | − | − | − | − | NP | 25% |
| 5 | + | − | + | − | − | weakly+ | − | − | 5% |
| 6 | + | + | − | − | + | NP | − | NP | 10% |
| 7 | + | + | − | + | weakly+ | − | weakly+ | − | 25% |
| 8 | + | + | + | − | − | − | − | − | 5% |
| 9 | + | + | NP | + | NP | NP | weakly+ | − | 30% |
| 10 | − | + | + | + | NP | − | NP | NP | 10% |
| 11 | + | − | + | − | + | − | − | NP | 15% |
| 12 | + | − | + | + | NP | − | NP | weakly+ | 8% |

**Notes.**

NP, Not performe.

years later, the left side of the face became swollen again, though without tenderness. Imaging examination revealed tumor recurrence, with invasion of the posterior wall of the maxillary sinus, pterygopalatine fossa, infraorbital wall, and parts of the inner wall. Therefore, a second surgery with a nasal lateral incision and a maxillectomy was performed. In consideration of the recurrence of the tumor and the scope of the disease, postoperative hormonal therapy and radiotherapy were administered. The tumor recurred 11 months later and was treated with combined radiotherapy and chemotherapy. At the last follow-up, the patient was still living with the disease.

The patient with nasopharyngeal IMT also underwent two surgeries. The first procedure was an endoscopic oral-nasal tumor resection, followed by adjuvant radiotherapy. Eighteen

Liu et al. (2025), *PeerJ*, DOI 10.7717/peerj.19315

**Table 3  Treatment and follow-up results of 12 HNIMT patients.**

| Case | Site | Surgical mode | Adjuvant therapy | Recurrence/ metastasis | Treatment after recurrence | Follow-up time | Status at last follow-up |
|---|---|---|---|---|---|---|---|
| 1 | Left nasal cavity | Resection of left nasal cavity mass under nasal endoscope | No | No | No | 36 months | Disease-free survival |
| 2 | Right nasal cavity and paranasal sinus | Resection of right nasal cavity and paranasal sinus mass under nasal endoscope | Yes (radiotherapy) | No | No | 48 months | Disease-free survival |
| 3 | Right parotid gland | Right superficial parotidectomy and mass excision | No | Yes | Right parotidectomy | 92 months | Disease-free survival |
| 4 | Left nasal cavity and paranasal sinus | Subtotal resection of left maxilla and nasal cavity mass under lateral nasal incision | No | Yes | Resection of left maxilla under lateral nasal incision + hormone + chemoradiotherapy | 55 months | With disease survival |
| 5 | Larynx | Laryngofissure and resection of right vocal cord mass | No | No | No | 24 months | Disease-free survival |
| 6 | Tongue | Resection of left tongue mass | No | No | No | 15 months | Disease-free survival |
| 7 | Nasopharynx | Combined oral and nasal approach for mass resection of nasopharynx under endoscope | Yes (radiotherapy) | Yes | Surgery + hormone + chemotherapy + ALK inhibitor | 44 months | With disease survival |
| 8 | Right submandibular gland | Right submandibular gland and mass excision | No | No | No | 16 months | Disease-free survival |

**Table 3** (*continued*)

| Case | Site | Surgical mode | Adjuvant therapy | Recurrence/ metastasis | Treatment after recurrence | Follow-up time | Status at last follow-up |
|------|------|---------------|------------------|------------------------|----------------------------|----------------|--------------------------|
| 9 | Subglottic en-dotracheal | Resection of sub-glottic endotracheal mass under endo-scope | No | Yes | Resection of sub-glottic endotra-cheal mass under endoscope | 18 months | Disease-free survival |
| 10 | Right neck | Extended excision of right neck mass | No | No | No | 10 months | Disease-free survival |
| 11 | Right gingiva | Extended resection of right buccal and gingival masses + partial mandibulec-tomy + transplanta-tion of anterolateral femoral skin flap + temporary tra-cheotomy | No | No | No | 30 months | Disease-free survival |
| 12 | Larynx | Extended resection of vocal cord mass under endoscope | Yes (radiotherapy) | No | No | 11 months | Non-neoplastic death |

months later, the tumor recurred, and endoscopic surgery was performed again, followed by preoperative oral steroid therapy for three weeks and postoperative hormone therapy for two months. However, the tumor recurred 13 months after the second surgery. Considering the tumor's location, adjuvant chemotherapy and the ALK inhibitor Crizotinib were initiated, and, as of the last follow-up, the patient was still living with the disease.

The patient with subglottic intratracheal IMT, a 7-year-old child, underwent two endoscopic operations. Upon admission, angiography of the tumor was performed, which showed a rich blood supply to the tumor. Embolization of the tumor feeding vessels was then performed, followed by a tracheotomy and endoscopic subglottic tumor resection. Two months later, an irregular bulge with rough mucosa appeared in the operation area, and the lumen was narrowed by about 30%. Endoscopic holmium laser thermal ablation was performed again, and the patient recovered well with a disease-free survival status.

All 12 patients were followed up until August 2024. At the final follow-up, nine patients were disease-free, two patients were living with the disease, and one patient had died.

## DISCUSSION

Inflammatory myofibroblastic tumor (IMT) is a rare mesenchymal tumor primarily composed of differentiated myofibroblastic spindle cells and is accompanied by significant infiltration of inflammatory cells including plasma cells, lymphocytes, and eosinophils (*Gros et al., 2022*). In the 2020 World Health Organization classification of soft tissue and bone tumors, IMT is defined as a low-grade malignant or borderline tumor with the potential for recurrence or metastasis (*Sbaraglia, Bellan & Dei Tos, 2021*). Prior to being formally recognized as IMT, it was often referred to by other names such as inflammatory pseudotumor, plasma cell granuloma, benign myofibroblastic tumor, xanthogranuloma, fibrous histiocytoma, and inflammatory myofibroblastic hyperplasia (*Bashir, Al Sohaibani & Al-Rikabi, 2018*; *Khatri et al., 2018*). The cause of this disease and its detailed pathogenesis have not yet been elucidated. Some researchers believe that various factors, including trauma, inflammation, autoimmune diseases, surgery, viral infections, and abnormal expression or mutations of the anaplastic lymphoma kinase (*ALK*) gene, may contribute to the occurrence and development of IMT (*Taiymi et al., 2023*; *Song & Zhu, 2021*; *Wang et al., 2023a*; *Wang et al., 2023b*). The pathophysiological mechanism of IMT, caused by the above factors, may be uncontrolled proliferation of myofibroblasts, which in turn leads to the formation of tumor lesions (*Liu et al., 2023*; *Shi et al., 2022*). Some researchers have also reported cases of IMT being discovered during follow-up visits after treatment for an initial primary tumor (*Shen et al., 2024*). In the current study, one patient with laryngeal IMT had previously undergone radiotherapy for upper esophageal squamous cell carcinoma at the hospital two years earlier. Although the esophageal lesions had partially undergone remission, laryngeal IMT was diagnosed during the follow-up. Whether there is a causal relationship between radiotherapy history and laryngeal IMT remains uncertain and requires further research for confirmation.

HNIMT can occur at any age and usually has two peak incidence periods; the first peak incidence period is before the age of 20 and the second is between 50 and 60 years. HNIMT

occurrence rates do not significantly differ between genders (*Nakano, 2023*; *Casanova et al., 2020*). In this study, patient age ranged from 7 to 72 years old, with a median age of approximately 48 years and a male to female ratio of approximately 1.4:1, which is consistent with previous reports. Although IMT can occur throughout the body, it is most commonly found in the lungs, followed by the mesentery, omentum, abdominal organs, retroperitoneal space, and other locations (*Xu et al., 2022*; *Siemion et al., 2022*). IMT occurring in the head and neck is more rare, accounting for only 5% of all IMT cases and 14%–18% of extrapulmonary IMT (*Li et al., 2020*; *Raitio & Losty, 2024*; *Zhang et al., 2020a*; *Zhang et al., 2020b*). The most frequent site of HNIMT is the orbit, but it can also occur in the maxillary sinus, nasopharynx, larynx, pharynx, salivary glands, thyroid, and other regions (*Baranov & Hornick, 2020*; *Patil et al., 2022*). Common signs and symptoms of nasal/sinus IMT include nasal congestion, epistaxis, headache, maxillofacial mass or swelling, and numbness (*Casanova et al., 2020*; *Raitio & Losty, 2024*). Orbital IMT may present with epiphora, eye swelling, and exophthalmos, while laryngeal IMT is typically accompanied by hoarseness, dysphonia, and dyspnea (*Kaytez, Kavuzlu & Oguz, 2021*; *Nair et al., 2020*). IMT of the salivary glands and thyroid mainly presents as a painless, slowly growing mass, although some patients may experience discomfort due to the compression of nearby structures including the trachea and esophagus (*Zhang & Liu, 2023*; *Shi et al., 2022*). Approximately 15%–30% of IMT patients may experience systemic symptoms including unexplained fever, weight loss, anemia, thrombocytosis, polyclonal hyperglobulinemia, and elevated erythrocyte sedimentation rate (ESR), procalcitonin (PCT), and c-reactive protein (CRP) levels (*Kerr et al., 2021*). In this study, the main clinical manifestations were also non-specific. One patient with nasal IMT had elevated levels of white blood cell counts, CRP, and PCT, along with an intermittent low-grade fever prior to surgery. After surgery, the fever symptoms disappeared, and levels of white blood cells, CRP, and PCT returned to normal.

Ultrasonography is the most commonly used initial screening tool for head and neck tumors and can determine the location, size, blood flow, and internal echo of a tumor (*Shi et al., 2022*). Endoscopy allows for direct visualization of the lesions, and electronic staining technology can help detect smaller lesions. CT and MRI have obvious advantages over ultrasound in evaluating the size and extent of tumors and their relationship with surrounding tissues (*Wang et al., 2023a*; *Wang et al., 2023b*). The various manifestations of HNIMT observed through imaging are diverse, and are related to the proportion of fibrosis in the lesion, inflammatory cell infiltration, and the stage of disease development. However, due to the lack of specific imaging features, HNIMT often appears as a solid mass, making it challenging to reach a clinical diagnosis based solely on imaging (*Yamamura et al., 2021*). Using ultrasonography, HNIMT typically presents as a hypoechoic solid mass with homogeneous or heterogeneous internal echo, well-defined or infiltrative margins, and fluid echoes in some lesions. Color Doppler Flow Imaging (CDFI) usually shows minimal blood flow signals in the mass (*Surabhi et al., 2016*). Endoscopy is mainly used in the examination of nasal and paranasal sinuses, larynx, and pharynx. IMT of the larynx often presents as a smooth surface, polypoid, or pedunculated bulge mass (*Tay & Balakrishnan, 2016*). On a CT scan, HNIMT usually appears as an expansive soft tissue

mass, either round or lobulated, with varying degrees of enhancement. The solid part of the lesion shows progressive enhancement, and some lesions may exhibit non-enhanced cystic necrosis or calcification (*Tan et al., 2017*). Due to the restrictive nature of the surrounding bone structure, the growth space of lesions in the head and neck is limited, allowing these lesions to easily invade the surrounding soft tissue and bone structure. This often results in local bone destruction and invasive growth, making it difficult to differentiate IMT from other malignant tumors when using imaging techniques (*Kandukuri & Phatak, 2016*). Compared to CT, MRI is more effective in revealing the relationship between IMT and nearby critical structures. Most lesions present as irregularly shaped nodules with unclear boundaries, with T1WI showing predominantly low signals and T2WI showing uneven, slightly high signals. Under enhanced MRI scans, tumors typically demonstrate moderate or strong enhancement, which may be due to the fibrous components and necrotic tissue within the lesion (*Buzan et al., 2020*; *Zeng et al., 2018*). The value of PET-CT in the diagnosis of HNIMT is somewhat limited due to its neoplastic and inflammatory features (*Manning et al., 2018*). Fludeoxyglucose-18 (FDG) uptake in HNIMT is highly variable, likely due to the number of tumor cells and the proportion and activity of inflammatory cells. Additionally, both HNIMT and malignant tumors can show increased FDG uptake (*Budylev et al., 2022*). Previous reports have documented cases where IMT was misdiagnosed as lymphoma based on PET-CT findings (*Ma et al., 2018*). Despite these limitations, PET-CT is helpful in the detection of primary tumors, local recurrence, distant metastasis, and assessment of treatment response (*Vounckx et al., 2024*; *Parker et al., 2024*). In the current study, patients received tailored imaging assessments based on the specific site of their lesions. The findings indicate that, although the imaging examination of HNIMT may suggest the possibility of malignant lesions, it is still difficult to make a definite diagnosis using imaging examination alone prior to surgery.

Since the discovery of IMT, the nomenclature and definition of this disease have evolved significantly. The term "IMT" has gradually been recognized by experts and scholars, and it is currently classified as a true tumor (*Theilen et al., 2018*; *Song & Zhu, 2021*). The diagnosis of HNIMT still relies on histopathology and immunohistochemical (IHC) staining (*Flores, De La Garza & Santillan-Gomez, 2021*). IMT is primarily comprised of spindle-shaped myofibroblastic cells and infiltrating inflammatory cells, often accompanied by fibrosis, hyaline degeneration, calcification, or necrosis. The inflammatory component can vary and usually includes plasma cells, lymphocytes, eosinophils, and neutrophils (*Wang et al., 2021*; *Gros et al., 2022*). Based on the proportion and distribution of various cells in the tumor tissue, IMT can be classified into three subtypes: mucinous/vascular type, spindle-cell dense type, and fibrous type (*Strianese et al., 2018*). Although all three histological subtypes can be present in the same lesion, usually one subtype predominates (*Choi & Ro, 2021*). IHC plays an important role in the diagnosis of HNIMT, as it helps determine the immunophenotype of myofibroblasts and rules out other diagnoses (*Cerier, Beal & Dillhoff, 2018*). In IMT, smooth muscle actin (SMA) is expressed in about 80%–90% of spindle cells, and muscle specific actin (MSA) and Desmin are found in approximately 60%–70% of cases, usually showing focal or diffuse positive expression (*Yang et al., 2021*; *Mittal et al., 2021*; *Gu & Chen, 2021*; *Elktaibi et al., 2020*). There is an unusually strong

and diffuse positive expression of vimentin (*Alan et al., 2020*; *Yun et al., 2020*; *Chang et al., 2019*), whereas S-100, CD34, CD117, CD21, and CD23 are usually not expressed. Therefore, the expression of Vimentin, SMA, and Desmin, and the absence of S-100 expression, constitutes a relatively high specificity for the diagnosis of IMT (*Parra-Herran, 2021*; *Wang et al., 2019*). Compared with other immunohistochemical markers, *ALK* has a unique significance in the diagnosis of HNIMT. *ALK* has been considered as a specific diagnostic marker and driver gene for IMT (*Mohammad et al., 2018*). Recent studies have confirmed that approximately 50%–70% of IMT patients have *ALK* gene rearrangement on chromosome 2p23, resulting in abnormal ALK expression (*Siemion et al., 2022*; *Parra-Herran, 2021*). *ALK* gene rearrangement can fuse with various partner genes, including *TPM3/4*, *ATIC*, *FN1*, *CLTC*, *GCC2*, *TFG*, *TNS1*, *THBS1*, and *DCTN1*, which may contribute to the tumor's development (*He et al., 2022*; *Collins et al., 2022*; *Son et al., 2023*; *Kuisma et al., 2022*; *Haimes et al., 2017*). In *ALK*-negative IMT, *ROS1* and *NTRK3* gene rearrangements are the most common mutations, occurring in 5%–15% of cases, while *RET* and *PDGFRB* rearrangements are relatively rare (*Takahashi et al., 2018*; *Huang et al., 2022*; *Yamamoto et al., 2016*; *Cheek et al., 2020*; *Comandini et al., 2021*). Some researchers recommend FISH analysis to assess ALK fusion status if IHC tests positive for ALK (*Chang et al., 2019*). In addition to IHC and FISH, next-generation sequencing (NGS), a more comprehensive molecular analysis, is another valuable tool for exploring the molecular mechanisms and genetic characteristics of IMT (*Racanelli et al., 2020*). In the current study, HE staining showed loose spindle-shaped and short spindle-shaped myofibroblastic proliferation, mild dysplasia, myxoid degeneration, hyalinization, and collagen formation in the stroma, accompanied by varying numbers of inflammatory cells including lymphocytes and plasma cells. IHC staining analysis showed that SMA, VIM, and ALK were positive in 83.3%, 66.7%, and 54.5% of cases, respectively. FISH was performed in two cases. Based on the typical histological and IHC features, it is not difficult to diagnose IMT.

During the development of HNIMT, a small number of cases may undergo malignant transformation, during which the tumor loses its original differentiation and becomes more aggressive. In cases of high-grade transformation, typical spindle cells may evolve into polygonal or round shapes, showing increased mitotic activity, cellular atypia, vesicular nuclei, large nucleoli, more frequent mitotic figures, and necrosis (*Chen et al., 2022*). As a borderline tumor, approximately 8%–18% of IMT cases may undergo malignant transformation during disease progression; this may require multiple biopsies, especially for recurrent or metastatic HNIMT (*Goyal et al., 2022*). In these cases, the confirmation of diagnosis by IHC is crucial (*Tao, Zhou & Zhou, 2015*). In the current study, four patients experienced recurrence after surgery, and all of these patients underwent subsequent surgical treatment. Pathological examination of specimens did not reveal malignant transformation, which may be attributable to the small number of cases included in our study.

As a borderline tumor, HNIMT has the potential for recurrence and distant metastasis. Therefore, it should neither be treated with the extensive resections used for malignant tumors, which carry risk of trauma, nor with the *in situ* resection used for benign lesions,

which could increase postoperative complications. To ensure complete resection, important structures should be preserved as much as possible to reduce functional damage (*Zhao et al., 2020*; *Mahajan et al., 2021*). According to the recommendations of the European Society for Medical Oncology (ESMO) guidelines and other studies, surgery remains the standard treatment for IMT. Surgery not only provides a definitive pathological diagnosis and symptom relief, but also effectively treats the tumor (*Vounckx et al., 2024*). An appropriate surgical approach should be selected based on the location, size, and extent of the tumor to ensure a successful radical operation (RO; *Gros et al., 2022*). Performing a frozen pathology examination during surgery is also recommended. The specific scope of tumor resection can then be determined based on the results of the frozen pathology examination (*Yuan, Fan & Xu, 2023*). Previous research found that most patients were cured after complete resection (*Siemion et al., 2022*). Studies by *Sagar, Jimenez & Shannon (2018)* and *Iwai et al. (2023)* have shown that the prognosis after RO resection is favorable, with a five-year survival rate of approximately 91%. The recurrence rate of IMT after RO resection varies depending on the location of the lesion. Pulmonary IMT has a recurrence rate of approximately 2%, while extrapulmonary IMT has a recurrence rate of around 25% (*Casanova et al., 2020*). In the head and neck region, IMT in the nasal cavity and sinuses tends to be more invasive and has a higher recurrence rate than IMT in other locations, while IMT in the salivary glands and thyroid has a lower recurrence rate (*Mahajan et al., 2021*). For patients with incomplete surgical resection, postoperative recurrence, or histological evidence of malignant transformation, comprehensive treatments such as chemotherapy, radiotherapy, or ALK inhibitors should be considered (*Casanova et al., 2020*). Studies on different chemotherapy regimens, including anthracyclines and vinorelbine/vinblastine, show an overall response rate (ORR) of 50%–64%, with some cases achieving long-term disease control (*Casanova et al., 2020*; *Baldi et al., 2020*; *Thompson, 2021*). Radiotherapy is rarely used in IMT and is mostly reported in case studies, with a lack of large-scale research to confirm its clinical efficacy (*Hou et al., 2020*). A study by *Peng et al. (2022)* revealed the potential benefit of adjuvant radiotherapy for HNIMT patients with malignant transformation. In this study, 45 patients underwent radical surgical resection, with 20 receiving adjuvant radiotherapy after surgery. The results showed that, while postoperative radiotherapy did not benefit the entire group of patients, it significantly improved overall survival in patients with malignant transformation. *Zhu et al. (2018)* and *Biswas et al. (2020)* reported a case where radiotherapy combined with steroid hormones was used to treat maxillary sinus IMT, resulting in a complete clinical and radiological response. In some cases, non-steroidal anti-inflammatory drugs (NSAIDs) or steroid hormones may reduce the local inflammatory response and decrease tumor size, although the mechanism by which this occurs remains unclear and requires further study (*Liu et al., 2023*; *Foster et al., 2021*). With the increased understanding of IMT pathology, neoadjuvant treatment of HNIMT has been gradually introduced into clinical practice. Neoadjuvant therapy may offer inoperable patients a chance for surgical resection by reducing tumor size and facilitating RO resection while preserving organ function, particularly in patients with complex head and neck anatomy (*Nagumo et al., 2018*; *Trahair et al., 2019*; *Comandini et al., 2021*).

Genetic abnormalities play a critical role in both diagnosing IMT and serving as key therapeutic targets. The expression of ALK in HNIMT is the foundation for personalized, molecularly targeted therapy (*Chmiel et al., 2024*; *Xu et al., 2022*). Crizotinib, a tyrosine kinase inhibitor targeting ALK, MET, ROS1, and RON, is recommended by the National Comprehensive Cancer Network (NCCN) as a treatment option for ALK-positive IMT (*Von Mehren et al., 2022*; *Trahair et al., 2019*; *Hunt et al., 2023*). For advanced, unresectable, recurrent/metastatic, or refractory ALK-positive IMTs, targeted therapy has become a critical component of treatment, with most cases resulting in favorable outcomes and manageable side effects (*Theilen et al., 2018*; *Siemion et al., 2022*; *Liu et al., 2023*; *Gros et al., 2022*; *Fischer et al., 2021*). In cases where resistance to crizotinib develops, second-generation ALK inhibitors such as ceritinib and alectinib (*Saiki et al., 2017*; *Mansfield et al., 2016*) or third-generation ALK inhibitors such as lorlatinib (*Comandini et al., 2021*; *Mai et al., 2019*) can be considered. ALK-negative patients generally face a higher risk of distant metastasis and a poorer prognosis (*Chavez & Hoffman, 2013*; *Liu et al., 2019*). For such cases, previous research has suggested the efficacy of non-steroidal anti-inflammatory drugs (NSAIDs; *Chavez & Hoffman, 2013*) and small-molecule tyrosine kinase inhibitors (*Liu et al., 2019*), although these reports are limited to individual cases. Mutations in *ROS1*, *NTRK*, *ETV6*, *RET*, *NTRK*, *PDGFR-β*, and other genes have also been identified in ALK-negative patients (*Yamamoto et al., 2016*; *Debonis et al., 2021*; *Chang et al., 2019*). For ALK-positive IMTs, neoadjuvant therapy with ALK inhibitors prior to surgery may reduce surgical risk for larger, multi-focal, or difficult-to-resection tumors (*Zhang et al., 2020a*; *Zhang et al., 2020b*).

After radical resection, the overall prognosis of IMT is favorable, the recurrence rate varies with the location of the lesion, and the occurrence of distant metastasis is relatively rare (*Casanova et al., 2020*; *Raitio & Losty, 2024*; *Bettach et al., 2022*; *Siemion et al., 2022*). HNIMT generally has a poorer prognosis compared to IMT in other sites, though it is unclear whether this is due to the aggressive biological behavior of the tumor or the more complex anatomy of the head and neck region (*Casanova et al., 2020*; *Pandya & Bhatt, 2021*; *Zhang et al., 2020a*; *Zhang et al., 2020b*). The local recurrence rate of HNIMT is approximately 10%–37%, which may be related to various factors such as surgical margins, tumor location, and tumor size (*Shi et al., 2022*; *Siddiqui et al., 2024*; *Gros et al., 2022*). Multiple studies have identified surgical margin status as a crucial predictor of local recurrence and overall survival (*Kurien et al., 2021*; *Serret Miralles et al., 2021*; *Peng et al., 2022*; *Parker et al., 2024*). In the head and neck, symptoms of sinonasal IMT usually appear after the mass occupies almost the entire region, most likely involving the orbit, pterygopalatine fossa, and skull base. The complexity of these important organs and the structure of the sinonasal cavity limit complete surgical resection, so HNIMT's high recurrence rate may be related to positive surgical margins (*Da et al., 2021*). Other factors that may affect prognosis include tumor size, pseudo-encapsulation, intralesional necrosis, and the Ki-67 label index (*Han et al., 2022*). The relationship between ALK expression and prognosis of IMT remains debatable. Some studies suggest that the expression of ALK is associated with a better prognosis (*Yorke et al., 2023*; *Arbour & Riely, 2019*). Although ALK overexpression may be one cause of tumors, ALK positivity also provides

the possibility of targeted therapy (*Arbour & Riely, 2019*). However, other studies believe that ALK-positive IMT carries a higher risk of local recurrence, while ALK-negative IMT tends to be more aggressive and prone to distant metastasis (*Hou et al., 2020*; *Siemion et al., 2022*; *Domínguez-Massa et al., 2023*). The optimal treatment regimen for recurrent HNIMT remains unclear. If only local recurrence occurs and the patient is in good physical condition, reoperation is still recommended (*Preobrazhenskaya et al., 2020*). For advanced HNIMT where surgery is no longer an option, individualized comprehensive treatment needs to be considered.

In the current study, four patients experienced recurrence after the first operation, including three recurrence cases with a special anatomical location (nasal cavity and paranasal sinus, nasopharynx, subglottic trachea) and one recurrence case located in the parotid gland. The recurrence of tumors may be related to the special anatomic location. In the cases of IMT within the nasal cavity, paranasal sinuses, and nasopharynx, symptoms typically manifest once the mass has nearly filled the entire space, often encroaching upon the orbit, pterygopalatine fossa, and skull base. The involvement of these critical structures, coupled with the intricate architecture of the nasal cavity and paranasal sinuses, constrains the possibility of a complete surgical resection, thereby increasing the likelihood of postoperative relapse. Although endoscopic surgery can better preserve laryngeal function and accelerate postoperative rehabilitation, it can also result in incomplete resection of the base of the lesion, causing local recurrence. In the current study, subglottic intratracheal IMT recurred after the initial operation and underwent extended resection. At the final follow-up, no recurrence has been observed. Therefore, for HNIMT, it is recommended that physicians preserve the relevant functions on the premise of complete tumor resection and give postoperative adjuvant treatment if necessary. One case of parotid IMT underwent resection of the superficial lobe and tumor of parotid gland during the initial operation. The postoperative pathology showed that the tumor lacked a capsule and had infiltrated the parotid and surrounding muscle tissues. Upon recurrence, a more extensive resection (total parotidectomy) was performed, leading to a favorable postoperative recovery and, at the final follow-up, there had been no observed recurrence. The initial recurrence seen in this case may be attributed to the inadequate scope of the initial surgical intervention.

There are limitations to the current study. First, the low incidence of HNIMT and the characteristics of single-center, retrospective studies make it difficult to eliminate selection bias. Second, given the rarity of HNIMT and the difficulty in collecting cases, we chose a retrospective case series study, which lacks a control group and assessment of exposure-outcome relationships.Third,the lack of a standardized treatment regimen for HNIMT may impact the assessment of its effectiveness in improving survival rates. In the future, multi-center, larger-sample prospective studies are needed , and researchers should consider case-control or cohort designs to better evaluate HNIMT's risk factors, treatment efficacy, and prognostic factors.

## CONCLUSION

HNIMT is a rare low-grade malignancy with a relatively favorable prognosis. However, achieving a definitive diagnosis may be difficult due to the lack of specific preoperative

symptoms and imaging findings. Histopathological analysis and immunohistochemistry techniques play crucial roles in the diagnosis of HNIMT. Surgical resection is the preferred treatment for HNIMT, particularly for tumors with well-defined margins. Achieving complete surgical removal often results in a favorable prognosis. For patients who have difficulties in undergoing surgery, an alternative strategy is to shrink the tumor through preoperative radiotherapy, hormone therapy, and other treatments, and then achieve complete resection. In recent years, targeted therapy based on gene mutations in IMT has been used. In particular, targeted therapy for ALK mutations has provided new therapeutic strategies for patients, and related ALK inhibitors have also been approved for clinical use. As genomics and genetic sequencing technologies advance, personalized medicine is poised to become a significant component of HNIMT treatment. Whether adjuvant therapy is required after surgery depends on the surgical margin, ALK gene mutation status, tumor size and location, patient age, and tumor stage. Although it is widely accepted that IMT is a low-grade malignant or borderline tumor, the potential for recurrence and metastasis necessitates close follow-up after treatment.

### Funding
The authors received no funding for this work.

### Competing Interests
The authors declare there are no competing interests.

### Author Contributions
- Feng Liu conceived and designed the experiments, performed the experiments, authored or reviewed drafts of the article, and approved the final draft.
- Yanchao Qin conceived and designed the experiments, authored or reviewed drafts of the article, and approved the final draft.
- Zhiwei Zhang conceived and designed the experiments, authored or reviewed drafts of the article, and approved the final draft.
- Mengru Li performed the experiments, analyzed the data, prepared figures and/or tables, and approved the final draft.
- Bowei Feng performed the experiments, prepared figures and/or tables, authored or reviewed drafts of the article, and approved the final draft.
- Wei Ding analyzed the data, authored or reviewed drafts of the article, and approved the final draft.
- Shubin Dong analyzed the data, prepared figures and/or tables, and approved the final draft.

### Human Ethics
The following information was supplied relating to ethical approvals (i.e., approving body and any reference numbers):

This study has received approval from the Medical Ethics Committee of Shanxi Cancer Hospital (Ethics number: KY2023231).

## Data Availability

The raw data are available in the Supplementary File.

## Supplemental Information

Supplemental information for this article can be found online at http://dx.doi.org/10.7717/peerj.19315#supplemental-information.

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
