# Peer review of "Therapeutic strategy and prognostic analysis of inflammatory myofibroblastic tumor in the head and neck: a retrospective study"

_PeerJ, doi:10.7717/peerj.19315_

## Round 0.1 · original submission · Major Revisions

The authors are requested to carefully revise the manuscript and answer the questions raised by the reviewers.

·

Basic reporting

Comments to the Author

This paper demonstrates the progress in the surgical resection of head and neck inflammatory myofibroblastic tumor (HNIMT), and the determination of the need for adjuvant therapy. The findings suggest a higher probability of specific preoperative symptoms and imaging findings.

However, some figures lack clarity and contain minor flaws, while certain sections require revisions for improved wording and grammar.

1. In line 908, when describing the special imaging techniques of NBI and IPCL, you may consider adding their full names to help readers understand. It would be beneficial to include "Narrow Band Imaging (NBI)" and "Intraepithelial Papillary Capillary Loop (IPCL)" for better clarity.

2. In lines 175-177, "According to differences in cell proportion and distribution, six cases were classified into three main types: mucinous/vascular type, spindle-cell dense type, and fibrous type." If you have characteristic macroscopic views of these six cases that illustrate their types, it would help readers understand what you intend to convey.

Experimental design

Regarding the figures, as mentioned for figure 1, the left parotid gland has unclear boundaries. It would be helpful to mark it clearly with an arrow or asterisk (*), and include a healthy tissue image for contrast. Similarly, for figure 2, the endoscopic finding of laryngeal IMT, it would be better to indicate the cyst-like mass site clearly.

Validity of the findings

In line 902, the figure 1 legend mentions the left parotid gland. However, table 3, which introduces all 12 cases, does not include a case with a left parotid gland tumor site. Could the authors provide a better explanation?

Additional comments

In conclusion, I think it is challenging to have a clear therapeutic strategy and prognostic analysis due to different tumor sites. Although you provide a forward-looking view to address this issue, it would be beneficial to summarize these points.

Additionally, lines 338-347 would be better placed in the results section. Similarly, lines 525-529 would be more appropriate in the results section.

Reviewer 2 ·

Basic reporting

Authors have described 12 cases of the inflammatory myofibroblastic tumor of the head and neck region (HNIMT) in an attempt to characterize them, improve our understanding of this disease, and provide valuable insights for the standardized diagnosis and management of HNIMT. This is relevant, but I have a few comments.

1. Title: myofibroblastoma - please stick to the standard terminology" myofibroblastic."

2. Line 96: From the second year onward, follow-up
97 appointments will be scheduled every six months.
Please rephrase it to reflect how the patients followed up.

3: Line 171: "sections" to be replaced by "surfaces"
Line 172: "short-shaped" What does that mean? Need better description

The discussion can be shortened by focusing on 12 cases and comparing them with earlier literature. Please emphasize your unique findings and explain why recurrences happened in your cases.

Experimental design

No comments

Validity of the findings

No comments

---

## Round 0.2 · Major Revisions

The authors are requested to carefully revise the manuscript and answer the questions raised by the reviewers.

·

Basic reporting

The manuscript presents a retrospective analysis of 12 cases of head and neck inflammatory myofibroblastic tumors (HNIMT), focusing on clinical features, treatment strategies, and prognosis. Given the rarity of HNIMT and the limited literature on its management, the study is relevant. However, it suffers from serious methodological and structural issues that significantly undermine its credibility and contribution to the field. Additionally, the manuscript lacks a strong hypothesis, a controlled experimental design, and rigorous statistical analysis, all necessary for an original article. Given its retrospective nature and focus on multiple cases, this study is better suited as a case series rather than an original research article.

Experimental design

1. In the retrospective study design, we have either a case-control study or a cohort study. In the former study design, it is a must to have a control group while the other study design assesses the relationship between exposure and an outcome. The study lacks a control group and it does not have any exposure and outcome relationship assessment.

2. The inclusion and exclusion criteria are not explicitly defined, raising concerns about selection bias.

3. The study includes only 12 patients, which is a very small sample size, even for a rare disease like HNIMT. While the authors acknowledge the rarity of the condition, the small sample size limits the generalizability of the findings. With inferential statistics, the results could be extrapolated to a broader population but the study lacks it as well.

4. The diagnostic criteria for IMTs are unclear, lacking details on histopathological and molecular markers.

5. Treatment protocols and follow-up periods are inconsistently described, leading to ambiguity in interpretation. The follow-up duration varies among patients, with some followed for only a short period. This variability could affect the assessment of recurrence and long-term outcomes.

Validity of the findings

1. The tables and figures are informative, but some of the legends are incomplete or unclear. For example, Figure 6 is labeled as "Pathological examination of larynx IMT," but the figure itself includes multiple stains (ALK, SMA, Desmin, etc.), and the legend does not explain what each panel represents.

2. The manuscript lacks any meaningful statistical analysis, which is a critical component of scientific research. Without statistical tests, it is difficult to assess the significance of the findings or compare outcomes between different treatment groups. This omission weakens the scientific rigor of the study.

Additional comments

The manuscript does not appear to present significantly novel findings compared to existing literature. The clinical features, treatment strategies, and outcomes described are largely consistent with what has been reported in previous studies. Without a unique contribution, the manuscript may not meet the journal's criteria for publication.

·

Basic reporting

The manuscript titled “Therapeutic strategy and prognostic analysis of inflammatory myofibroblastic tumor in the head and neck : a retrospective study” undertook a report of 12 cases inflammatory myofibroblastic tumor of the head and neck region (HNIMT) , the authors elaborated the tumors’ clinical, pathology, imaging manifestations, treatment and follow-up information,particular attention was paid to the surgical intervention and follow-up results of recurrence of HNIMT. This may enrich the literature of related fields and provide valuable insights for the diagnosis and management of HNIMT.
The writing is smooth and the sentences are easy to be understood.

Experimental design

A few tips:

1. The Results part in the ABSTRCT is too long and needs to be simplified
2. The specificity and accuracy of preoperative imaging diagnose of HNIMT are not satisfactory, I would suggest combined Figure 1-4 into one BIG figure with one ultrasound image, one endoscopic image, and 1-2 CT and MRI images, with detailed figure description including case number, age, lesion location and size of the HNIMT. For all measurements, an abnormal mass can be displayed, but there are always no specific signs for confirming the diagnose of HNIMT
3. Usually, for CT and MRI, we routinely express it as: image with contrast and/or image without contrast showed/demonstrated... Instead of CT examination/Enhanced scan/Plain scan present...
3. I suggest that a column be added to Table 3 to indicate the living status of the patient at the last follow-up
4. Descriptions of CT and MRI images should be written in the passive voice. For example, Line115-117: “The tumors mostly presented as solid nodules, and a few appeared as cystic-solid nodules with punctate blood flow signals within the lesions”
This can be written like: HMIMTs are mostly demonstrated as solid nodules on US images, and a few may be appeared as cystic-solid nodules with punctate blood flow signals within the lesions.
There are many similar expressions in the manuscript which would be understood more smoothly if some modifications were made

Validity of the findings

no comment

Additional comments

no comment

---

## Round 0.3 · accepted · Accept

After revisions, two of the three reviewers have agreed to accept the manuscript for publication. Although one reviewer suggested rejecting the manuscript and resubmitting it as a case series report, I believe the editorial board could consider the form of publication for this manuscript without rejection.

·

Basic reporting

Thank you for your detailed responses to the reviewers' comments and for your efforts in revising the manuscript. We appreciate the time and effort you have dedicated to addressing the concerns raised.

After careful consideration of the reviewers' feedback and your point-by-point responses, we acknowledge that your study contributes to the understanding of inflammatory myofibroblastic tumors in the head and neck (HNIMT). However, several critical concerns remain unresolved, particularly in terms of methodological rigor, statistical analysis, and novelty.

Key Issues Considered in the Decision:
Methodological Limitations – The study is a retrospective case series lacking a control group, exposure-outcome analysis, and robust statistical methods. While the authors acknowledge these limitations, the absence of statistical analysis significantly weakens the study's scientific rigor and generalizability.

Small Sample Size – With only 12 cases, the study lacks sufficient power to provide meaningful conclusions applicable to a broader population. While HNIMT is rare, future studies with a larger, multi-center cohort would be more impactful.

Lack of Novelty – The findings largely reiterate existing literature rather than providing significant new insights. The authors argue that the study is valuable for regional data representation, but the lack of unique contributions limits its suitability for publication in its current form.

Issues with Data Presentation – While revisions have improved clarity, concerns remain about figure labeling, the need for a more structured follow-up section, and the presentation of imaging findings.

Experimental design

Experimental Design:

While the study provides a retrospective analysis of 12 cases of head and neck inflammatory myofibroblastic tumors (HNIMT), several methodological limitations affect its robustness:

Study Design & Control Group: The study is structured as a retrospective case series rather than a cohort or case-control study. Without a control group or exposure-outcome analysis, the ability to draw meaningful comparative conclusions is limited. This reduces the scientific impact of the study in the context of evaluating treatment effectiveness and prognostic factors.

Inclusion & Exclusion Criteria: The authors have now provided clear inclusion and exclusion criteria. However, the initial lack of explicit criteria may have introduced selection bias. Future studies should outline these parameters more comprehensively in the methodology section to ensure reproducibility and transparency.


Recommendation:
While the study offers valuable clinical insights, it lacks the methodological rigor necessary for an original research article. A more structured design, inclusion of statistical analysis, and a larger sample size would enhance its scientific validity. Alternatively, the study could be restructured as a case series report with a focus on clinical observations rather than attempting to draw broader conclusions.

Validity of the findings

Sample Size & Statistical Analysis: The study includes only 12 cases, which is quite small even for a rare condition like HNIMT. While the authors acknowledge this limitation, the lack of statistical analysis further weakens the study’s rigor. Without statistical validation, the generalizability of findings is minimal. Incorporating inferential statistics or expanding the dataset through multi-center collaboration would significantly enhance the study’s credibility.

Diagnostic Criteria & Follow-Up: The manuscript has improved its description of histopathological and molecular markers for diagnosis. However, follow-up durations are inconsistent across cases, making it difficult to reliably assess recurrence rates and long-term outcomes. Standardizing follow-up protocols and ensuring a more uniform follow-up period would strengthen future analyses.

Additional comments

Manuscript Structure & Readability – The overall writing is clear, but certain sections could benefit from improved organization and concise phrasing. In particular, the abstract results section is too long and should be streamlined to highlight key findings more effectively.

Figures & Tables – The manuscript includes relevant tables and figures, but some figure legends remain unclear. For instance, Figure 6 lacks proper labeling for immunohistochemical markers. Ensuring that all images have detailed captions explaining their significance would improve clarity.

Clinical Relevance & Novelty – While the study contributes to the literature on HNIMT, it does not present substantially novel findings. The results align closely with previously reported clinical features and treatment approaches. The authors should further clarify how their study adds new insights beyond existing knowledge.

Ethical & Institutional Considerations – The study would benefit from a clearer statement on ethical approval and patient consent to confirm compliance with institutional and ethical guidelines.

Formatting & Consistency – Minor inconsistencies in numerical representation (e.g., ranges and units) and terminology usage should be reviewed for uniformity throughout the manuscript.

Reviewer 2 ·

Basic reporting

The manuscript is well-written and has improved significantly compared to the earlier version.

Experimental design

No comments

Validity of the findings

No comment

·

Basic reporting

No comment

Experimental design

No comment

Validity of the findings

No cnomment

Additional comments

Please classify this manuscript as a case series.